# Effectiveness of Internet- and Mobile-Based Cognitive Behavioral Therapy to Reduce Suicidal Ideation and Behaviors: Protocol for a Systematic Review and Meta-Analysis of Individual Participant Data

**DOI:** 10.3390/ijerph17145179

**Published:** 2020-07-17

**Authors:** Rebekka Büscher, Marie Beisemann, Philipp Doebler, Lena Steubl, Matthias Domhardt, Pim Cuijpers, Ad Kerkhof, Lasse B. Sander

**Affiliations:** 1Department of Rehabilitation Psychology and Psychotherapy, Albert-Ludwigs-University of Freiburg, 79106 Freiburg, Germany; lasse.sander@psychologie.uni-freiburg.de; 2Department of Statistics, TU Dortmund University, 44227 Dortmund, Germany; beisemann@statistik.tu-dortmund.de (M.B.); doebler@statistik.tu-dortmund.de (P.D.); 3Department of Clinical Psychology and Psychotherapy, Ulm University, 89069 Ulm, Germany; lena.steubl@uni-ulm.de (L.S.); matthias.domhardt@uni-ulm.de (M.D.); 4Department of Clinical, Neuro and Developmental Psychology, Amsterdam Public Health Research Institute, Vrije Universiteit Amsterdam, 1081 BT Amsterdam, The Netherlands; p.cuijpers@vu.nl (P.C.); ajfm.kerkhof@gmail.com (A.K.)

**Keywords:** individual participant data meta-analysis, review, suicidal ideation, suicide, internet-based, mobile-based, online, cognitive behavioral therapy, iCBT

## Abstract

Internet- and mobile-based cognitive behavioral therapy (iCBT) might reduce suicidal ideation. However, recent meta-analyses found small effect sizes, and it remains unclear whether specific subgroups of participants experience beneficial or harmful effects. This is the study protocol for an individual participant meta-analysis (IPD-MA) aiming to determine the effectiveness of iCBT on suicidal ideation and identify moderators. We will systematically search CENTRAL, PsycINFO, Embase, and Pubmed for randomized controlled trials examining guided or self-guided iCBT for suicidality. All types of control conditions are eligible. Participants experiencing suicidal ideation will be included irrespective of age, diagnoses, or co-interventions. We will conduct a one-stage IPD-MA with suicidal ideation as the primary outcome, using a continuous measure, reliable improvement and deterioration, and response rate. Moderator analyses will be performed on participant-, study-, and intervention-level. Two independent reviewers will assess risk of bias and the quality of evidence using Cochrane’s Risk of Bias Tool 2 and GRADE. This review was registered with OSF and is currently in progress. The IPD-MA will provide effect estimates while considering covariates and will offer novel insights into differential effects on a participant level. This will help to develop more effective, safe, and tailored digital treatment options for suicidal individuals.

## 1. Introduction

Worldwide, an estimated 800,000 people die by suicide every year [1]. Suicidal ideation and attempted suicide are far more frequent [1], with a worldwide lifetime prevalence of 9.2% and 2.7%, respectively [2]. Suicidal ideation and behaviors (SIB) can be enormously burdensome for individuals and their personal environment [1], and they entail vast economic costs for societies [3]. Despite suicide prevention efforts, suicide rates have increased by 60% in the last 45 years [4]. Hence, suicide prevention remains a worldwide public health challenge [1,5].

As SIB are highly complex phenomena, suicide prevention strategies follow a multidimensional approach and comprise a variety of interventions and target groups [6]. Examples of effective suicide prevention strategies are the restriction of lethal means (e.g., painkillers), school-based awareness programs, and (telephone-) counseling [7]. Since suicidality frequently occurs in the context of mental disorders, psychotherapy and other psychosocial interventions play a key role in suicide prevention [7]. Approaches such as cognitive behavioral therapy (CBT) and dialectical behavioral therapy (DBT) have been shown to be effective in reducing SIB [7,8,9,10]. The reduction of suicidal ideation is important regardless of the presence of suicidal behaviors, as it can be extremely burdensome for individuals [8]. In addition to that, it is a major risk factor for suicide attempts [11,12,13,14]. Even passive ideation, such as a wish to die, has been identified as a risk factor for death by suicide [15], although it remains widely unclear which individuals experiencing suicidal ideation will proceed to suicide attempts [16,17]. The timely reduction of suicidal ideation could be a highly effective suicide prevention strategy in the long term, since 60% of the transitions from suicidal ideation to suicidal behaviors occur within 12 months after onset [2].

While there are effective treatment approaches, many suicidal persons do not receive treatment [18]. Barriers to treatment include low perceived need, the wish to solve the problem by oneself, the perception of the problem as not that severe, perceived stigma, and structural barriers such as financial issues or low availability [18]. Internet- and mobile-based interventions have the potential to overcome many of those barriers by offering highly flexible, scalable and accessible self-help interventions, and an anonymous setting [19]. This approach might be especially acceptable for suicidal individuals, as suicidal ideation is particularly prevalent in online help seekers with common mental disorders [20]. Moreover, many internet- and mobile-based interventions are designed as self-help interventions that enable persons to solve the problem on their own [19].

Within the last two decades, vast research has documented the effectiveness of this new treatment approach for a variety of mental disorders [21,22,23,24,25,26]. Internet- and mobile-based interventions for SIB, most of them based on cognitive behavioral therapy (iCBT), have been developed and evaluated in recent years. Previous systematic reviews and meta-analyses found mixed results for their effectiveness. Some reviews suggest that they might be effective in reducing suicidal ideation [27,28,29,30,31], with effect sizes in meta-analyses of randomized controlled trials ranging from Hedges’ *g* = −0.23, 95% CI −0.35 to −0.11 [27] to *g* = −0.29, 95% CI −0.40 to −0.19 [28] at post-intervention. Other reviews, however, did not find significant positive effects on suicidal ideation [32,33,34]. Low treatment adherence further limits the confidence in the benefits of this new treatment approach for clinical practice [27,28,29]. 

These inconsistent findings might result from differences in study characteristics (e.g., type of control group, setting, recruitment strategy), participant characteristics (e.g., history of suicide attempts, severity of suicidal ideation), and intervention characteristics (e.g., level of human support, treatment duration). Moreover, even in those studies with positive results, the estimate of an overall effect size does not reveal beneficial or harmful effects at individual participant level. There might be subgroups of individuals who benefit more from the treatment than others.

The analysis of individual participant data (IPD) is a powerful tool to address the outlined limitations of previous meta-analyses. Offering several advantages over meta-analyses of aggregated data [35,36], individual participant data meta-analyses (IPD-MA) are considered to be the gold standard of evidence synthesis [37]. IPD-MA provide a more precise effect estimate by reducing statistical heterogeneity. Especially for the precise estimation of effects on participant level, such as subgroup effects or participant-level moderators as well as rare outcomes such as negative effects, it is insufficient to rely on aggregated data [38,39,40,41].

Hence, this is the study protocol for an IPD-MA that aims to:Determine the effectiveness of iCBT specifically targeting SIB on suicidal ideation;Assess the clinically relevant changes (reliable improvement and deterioration, response rate) regarding the effects on suicidal ideation;Identify effect moderators on participant level, intervention level, and study level;Examine treatment adherence and potential predictors for adherence.

## 2. Materials and Methods 

This review will be reported according to the Preferred Reporting Items for a Systematic Review and Meta-Analysis of Individual Participant Data (PRISMA-IPD) guidelines [42]. This protocol adheres to the PRISMA Protocols statement [43]. We registered the review with the OSF (https://osf.io/45tcd). Potential amendments to the protocol will be documented and detailed in the final report.

### 2.1. Eligibility Criteria

Trials will be selected based on the following PICOS criteria (see Table 1). We will include studies that were published in a peer-reviewed journal irrespective of language and publication date. Relevant articles published in a language that is not spoken by the authors will be translated.

#### 2.1.1. Participants

The population of interest consists of individuals experiencing suicidal ideation at baseline. We will include studies on suicidal individuals irrespective of age, psychiatric diagnoses, or co-interventions (e.g., medication).

#### 2.1.2. Interventions

We will include trials on stand-alone iCBT interventions (including third-wave approaches such as DBT or mindfulness-based interventions) that directly address SIB. The interventions must be delivered primarily in an internet- or mobile-based setting, i.e., they must be designed as online interventions that can be used self-reliantly. Interventions may involve additional human support, e.g., written or personal feedback by a therapist or e-coach. Furthermore, they may involve limited face-to-face contacts for assessments or when the iCBT intervention is accessed in an institution, e.g., a clinic. Blended concepts (i.e., face-to-face interventions as the primary delivery mode), interventions without suicide-specific treatment elements targeting other mental health disorders (such as depression), interventions for “gate-keepers” such as teachers or social workers, help-seeking interventions, and interventions targeting stigma threat will be excluded.

#### 2.1.3. Comparators

Control groups may consist of treatment as usual (TAU), another active or passive controls (e.g., attention control, placebo), no intervention, or consist of a wait-list group.

#### 2.1.4. Outcomes

Studies are eligible if they report a quantitative measure of suicidal ideation. If studies provide multiple measures of suicidal ideation, they will be prioritized as follows: (1) validated self-report instruments (e.g., Beck Scale for Suicidal Ideation [44]), (2) clinician ratings, and (3) single item analysis of other scales (e.g., PHQ-9 [45]). If there are several measures within the prioritized category, we will select the measure that is most commonly used across all included trials.

#### 2.1.5. Study Design

We will include randomized controlled trials.

### 2.2. Study Identification and Selection Process

We will perform systematic literature searches in the following databases: the Cochrane Central Register of Controlled Trials (CENTRAL), PsycINFO, Embase, and Pubmed. We have developed a broad search strategy (see Appendix A) to generate a database that will be updated on a regular basis and used for future analyses. The search strings include a variety of search terms, combining the topics of suicide, internet and e-health, and randomized controlled trials. We developed our search strategy incorporating search terms from previous conventional meta-analyses and reviews [27,34,46,47]. We performed a pilot testing of the search strategy using a validation set of eight relevant trials. 100% of these trials were identified using the developed search strings. After the removal of duplicates, two reviewers will independently screen the studies for eligibility in a hierarchical approach. In a first step, the researchers will screen titles and abstracts and remove ineligible trials. In a second step, potentially relevant full texts will be assessed for eligibility, and reasons for exclusion will be tracked. Potential discrepancies will be resolved in discussion with a third researcher. If the eligibility cannot be determined using the reported information, we will contact authors for clarification.

The systematic database searches will be complemented by backward and forward searches to identify eligible trials that could not be identified in the electronic database searches: First, we will screen reference lists of included trials and relevant reviews that were identified in the systematic database searches, and second, we will search for studies that cited included studies and relevant reviews using Web of Science. Following the electronic searches, we will consult all authors who participate in the IPD-MA (i.e., who provide data) to identify any additional eligible trials. The selection process will be reported in a PRISMA flowchart.

### 2.3. Data Collection and Data Items

We will initially contact first and senior authors via e-mail and invite them to provide raw data for this IPD-MA. Two authors of each eligible trial will be invited to become co-authors in subsequent publications that contain the data from their trial, under the condition that they meet general criteria for authorship. If the authors do not reply to our request within two weeks, we will send an e-mail to all authors of the respective publication. Reminders will be sent after 2 and 4 weeks, if necessary. If authors fail to answer or cannot provide their data, we will exclude the respective trial from the IPD-MA and perform an additional meta-analysis of aggregated data to examine differences between included and excluded trials. In order to minimize the efforts on the part of participating authors, we will accept raw data in all formats. Study authors will receive regular e-mail updates to facilitate the collaboration.

Two independent reviewers will independently extract and code relevant data items from the published articles. A third reviewer will help to reach consensus among the coders if necessary. We will use an investigator-developed data form that we will pilot and modify if necessary. After receiving the raw data from authors, we will perform a data-check with the published article, comparing means and standard deviations of the measure of suicidal ideation and other baseline variables. Potential inconsistencies or any questions concerning the datasets will be discussed with the study authors.

In addition to the data extracted from the articles, a harmonized version of the raw IPD will be prepared for each study separately. After this, we will include all harmonized IPD datasets in a final merged IPD-MA dataset that will be used for the analyses. We will include all randomized participants into the merged IPD database.

### 2.4. Risk of Bias Assessment

Potential bias will be evaluated by two independent reviewers. First, we will evaluate the risk of bias based on published reports using the revised version of the Cochrane Risk of Bias Tool, the RoB 2 [48]. Risk of bias will be assessed in the following domains: (1) bias arising from the randomization process, (2) bias due to deviations from intended interventions, (3) bias due to missing outcome data, (4) bias in measurement of the outcome, and (5) bias in selection of the reported result. In addition, the tool involves an overall judgment of risk of bias.

Second, we will assess potential bias based on IPD. To date, there is no standardized tool for this purpose. However, the use of IPD might lead to a judgment of risk of bias that differs from the one based on the published data and information. For example, when the risk of bias is rated high because the outcome has been selected from several measures, all assessed outcomes can be included in the IPD-MA, leading to a low risk of bias. Therefore, we will additionally perform IPD-specific risk of bias ratings by re-evaluating the domains of the RoB 2 based on the IPD and by assessing other potential sources of bias. This will include the attenuation of mean differences, correlations, and regression coefficients due to range restrictions, (un-) reliability and high or low variances, and biases due to sample decomposition (sub-clinical vs. clinical, admitted to clinic vs. ambulances, co-morbidities). In addition, we will prepare an ORBIT (Outcome Reporting Bias in Trials) matrix to distinguish any data that were not collected in the primary investigations from data that were collected and are/are not made available as IPD. The open source ORBIT Matrix Generator (http://outcome-reporting-bias.org/) will be used.

### 2.5. Individual Participant Data Meta-Analysis

We will perform a one-stage IPD-MA. If computational problems arise from the one-stage IPD-MA, we will perform the two-stage IPD-MA for all analyses instead. If there are computational problems in the two-stage IPD-MA or if we do not have access to IPD from all eligible trials, we will conduct a conventional meta-analysis based on aggregated data. The one-stage approach combines the IPD from all included trials simultaneously in a hierarchical model, whereas the two-stage approach computes effects estimates for each trial separately and performs a traditional meta-analysis in a second step [49]. The one-stage approach is commonly chosen as it uses all information stored in the data, produces less biased effect estimates, accounts for parameter correlation [50], and provides more sophisticated moderator analyses [38].

We will perform the IPD-MA using three transformations/indices of suicidal ideation. (1) The reliable change index per person (RC [51]) will be used to indicate whether a relevant improvement, no relevant change, or deterioration occurred (ordinal coding with three categories). In addition, we will use the (2) response rate with a binary coding, defined as a 50% reduction of the suicidal ideation score at baseline. Furthermore, we will calculate the IPD-MA for the (3) continuous measures of suicidal ideation. In case of different outcome measures, we will transform the measures of suicidal ideation into *z* scores within each study. For the RC (1), we will perform an ordinal regression in the one-stage IPD-MA, as it includes all three categories in one analysis (i.e., deterioration, no change, improvement). If there are computational problems such as insufficient observations in a category, we will collapse two categories into one and fit a logistic model instead. For the response rate (2), we will perform a logistic multi-level regression in the one-stage IPD-MA. In case of two-stage IPD-MA for (1) and (2), the log odds ratios will be pooled for each study using the inverse-variance method used in traditional meta-analyses. For (1), we will calculate the log odds ratio with the three categories collapsed in two. For the continuous measure (3), we will perform a multilevel mixed-effects linear regression with participants nested within studies (one-stage IPD-MA). In case of two-stage IPD-MA for (3), we will calculate Hedges’ g and standard errors for each trial separately and pool them across the trials.

To account for the hierarchical structure of the data (i.e., patients nested in studies), we are going to model a random intercept in each of the three one-stage IPD-MA models. The treatment effect can then be modeled as fixed (for a homogenous effect across studies, referred to as a fixed-effects or mixed-effects model in the meta-analytical literature) or as random (for a heterogeneous effect across studies, referred to as a random-effects model). These two models will be compared via appropriate means of model comparison (analogous to the Q-test) to test whether the homogeneity assumption is justified.

We have chosen the outlined procedure in order to display different aspects of effectiveness. Furthermore, each of the three different approaches creates problems when standing alone. The RC (1) and the response rate (2) will not use all the information contained in the data due to the binary coding of clinically relevant changes in symptom severity. The RC (1) will be determined for each instrument separately, limiting the comparability across trials with different measures of suicidal ideation. The response rate (2) is determined independently of the specific instrument. However, individuals with a higher baseline severity might reach a 50% reduction of the suicidal ideation score more easily due to methodological artifacts such as the regression to the mean. The continuous measure of suicidal ideation (3) will not account for differences in variation due to the z-standardization, and the overall effect size will not reveal whether the relative difference is clinically relevant. Therefore, the three complimentary approaches will be combined to provide a robust and comprehensive analysis of the effects on suicidal ideation.

Beyond the analyses of suicidal ideation, we will perform an IPD-MA with a continuous index of treatment adherence as the dependent variable, analogous to the procedure described for (3). Treatment adherence will be defined as the proportion of completed modules.

All analyses will be performed using an intent-to-treat approach (ITT), including all participants who have been randomized regardless of their adherence to treatment. We will account for missing data using multiple imputation using the predictive mean matching method as implemented in the R package mice [52]. Any variables that are missing entirely from one study will not be imputed in order to be conservative. All analyses will be conducted using R [53].

### 2.6. Sensitivity Analyses

Several sensitivity analyses will be conducted. First, to examine potential differences between participants who dropped out of the trial and those who completed the assessments, we will conduct a sensitivity analysis using data from participants who completed all assessments. Second, we will perform a sensitivity analysis for age groups ((1) interventions for youth vs. interventions for adults/general population and (2) excluding all participants <18 years), and third, for trials at low risk of bias. Fourth, we will additionally perform a logistic one-stage meta-analysis based on the reliable change index with the three categories (i.e., deterioration, no change, improvement) collapsed in two to ensure comparability with two-stage IPD-MA.

### 2.7. Moderators of Suicidal Ideation Severity

We will examine whether characteristics of participants, interventions, or studies moderate the effects on suicidal ideation. The moderator analyses will be performed in the one-stage IPD-MA. If feasible, the following pre-defined variables will be examined as potential effect moderators. We will examine the clinical variables baseline severity of suicidal ideation, history of suicide attempts, depressiveness, hopelessness, anxiety, as well as the sociodemographic variables age, sex, level of education, relationship status, employment status, and treatment history. Furthermore, we will examine the study-level moderators human support [25,54,55], treatment dose, type of control group, and risk of bias. Beyond the a priori defined variables, we will perform exploratory moderator analyses for the effects on suicidal ideation. Furthermore, we will perform exploratory analyses of predictors for treatment adherence.

### 2.8. Meta-Analysis of Aggregrated Data

We will perform a conventional meta-analysis using the aggregated data reported in the original publications if necessary. For continuous outcome measures in randomized controlled trials, standardized mean differences (Hedges’ g) will be calculated and pooled using a random effects model with REML estimation. We will calculate a between-group effect size using changes from baseline to control for potential baseline differences between groups [56]. The conventional meta-analysis will be used to examine potential differences between the trials that provided data for the IPD-MA and those that did not, using a subgroup analysis. For binary outcomes, log Odds Ratios (log-ORs) will be computed and pooled using REML-estimation.

Potential publication bias will be assessed using the visual inspection of a funnel plot [57] and Egger’s test [58]. The risk of bias assessment discussed further above also includes a check for outcome reporting bias.

### 2.9. Quality of Evidence

Two independent reviewers will assess the quality of evidence on outcome level using the Grading of Recommendations Assessment, Development and Evaluation (GRADE). We will evaluate the quality of evidence for the primary outcome in the following domains: risk of bias, inconsistency, indirectness of evidence, imprecision of the estimate, and publication bias. Potential discrepancies between the GRADE ratings will be resolved by a third researcher.

## 3. Discussion

Suicidal ideation is a highly prevalent phenomenon [59]. As many individuals who experience suicidal ideation will transit to suicidal behavior [2], safe and effective treatment options are required. There is some emerging evidence that iCBT for SIB might be effective in reducing suicidal ideation [27,28,29,34]. However, the confidence in the effectiveness of iCBT is limited because the effect sizes are small, there are low levels of treatment adherence, and there might be imprecisions in effect estimates. Furthermore, it remains unclear whether there are subgroups of participants who experience clinically relevant reductions of suicidal ideation or harmful effects. In addition, it remains unclear whether treatment adherence can be predicted by characteristics of participants or interventions. Thus, the present protocol outlines the methods of a planned study that will extend knowledge derived from previous meta-analyses [27,28,29,34] by analyzing individual participant data. Thereby, this IPD-MA will provide an effect estimate while controlling for participant-level covariates and offering novel insights into differential effects on participant level.

Aside from several strengths (e.g., IPD-MA as the gold standard approach in evidence synthesis, a rigorous methodological approach, analyses on an individual level), there are potential study limitations that should be considered. First, we might not be able to obtain all relevant datasets, which might introduce some bias. However, to detect a potential bias in this regard, we will additionally conduct a meta-analysis using aggregated data from all identified eligible trials. Second, there may be inconsistencies across included studies concerning reporting and measurement of variables, which might limit the synthesis of available data and moderator analyses.

## 4. Conclusions

Comprehensive evidence on iCBT for SIB, including beneficial and harmful effects and potential treatment moderators, is urgently needed and of high clinical relevance. The field of mental health is moving towards precision medicine, which aims to match patients to their indicated treatments, acknowledging that no single intervention can be expected to be best for every patient [60]. Therefore, the results of this review will contribute to providing effective iCBT interventions to those individuals who are likely to benefit from this treatment approach. This will add to worldwide suicide prevention efforts, helping to prevent the chronification of suicidal ideation, the transition to suicide attempts, and premature deaths by suicide.

## Figures and Tables

**Table 1 ijerph-17-05179-t001:** Eligibility criteria (PICOS).

Participants	Experiencing Suicidal Ideation at Baseline
Interventions	Interventions must specifically target suicidal thoughts or behaviors Based on cognitive behavioral therapy including third-wave approaches Delivered in an internet- or mobile-based setting Guided or self-guided interventions Exclusion: Blended concepts (primary delivery mode: face-to-face) Gatekeeper interventions Help-seeking interventions Interventions targeting stigma
Comparisons	TAU, another active/passive treatment, placebo, waitlist, no intervention
Outcomes	Quantitative measure of suicidal ideation
Study design	Randomized controlled trial

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
