# Peer review of "Effectiveness of Internet- and Mobile-Based Cognitive Behavioral Therapy to Reduce Suicidal Ideation and Behaviors: Protocol for a Systematic Review and Meta-Analysis of Individual Participant Data"

_ijerph, 2020, doi:10.3390/ijerph17145179_

Round 1
Reviewer 1 Report
Abstract says the work is in progress therefore it is not completed. There are no results therefore until the study is completed it is not possible to fully assess the work.
Author Response
We thank the reviewer for this comment. As the reviewer mentioned, this manuscript does indeed not include results. In the field of individual participant meta-analyses (IPD-MA), it is good practice to publish study protocols outlining the methods that will be used. This enhances the quality of the final analyses as it reduces sources of bias. Study protocols offer the chance to describe the procedures in detail compared to the results paper that have often limited space for the methods section due to restrictive word counts. Furthermore, an IPD-MA is a collaborative project. The published protocol in a special issue can contribute to obtain data from eligible trials that are currently in the publishing process, as authors in the field will be likely to read the review. Therefore, we would highly appreciate the publication of this manuscript ahead of conducting the analyses.
Examples for IPD-MA study protocols:
Breedvelt, J.J.F.; Warren, F.C.W.; Brouwer, M.E.B.; Karyotaki, E.; Kuyken, W.; Cuijpers, P.; van Oppen, P. van; Gilbody, S.; Bockting, C.L.H. Individual participant data (IPD) meta-analysis of psychological relapse prevention interventions versus control for patients in remission from depression: a protocol. BMJ Open 2020, 10, e034158.
Lin, J.; Scott, W.; Carpenter, L.; Norton, S.; Domhardt, M.; Baumeister, H.; Mccracken, L.M. Acceptance and commitment therapy for chronic pain : protocol of a systematic review and individual participant data meta-analysis. 2019, 8, 1–10.
Karyotaki, E.; Furukawa, T.A.; Efthimiou, O.; Riper, H.; Cuijpers, P. Guided or self-guided internet-based cognitive-behavioural therapy (iCBT) for depression? Study protocol of an individual participant data network meta-analysis. BMJ Open 2019, 9.
Reviewer 2 Report
The paper described protocol for individual participant data meta-analysis (IPD-MA) regarding the determination of the effectiveness of internet- and mobile-based cognitive behavioral therapy on suicidal ideation and effect moderators on participant level, intervention level, and study level as well
Reviewers described their extensive search for studies, such as searched databases or whether efforts were made to contact the authors, to judge the risk of publication bias. The authors assess the quality of the included studies and report their findings. To minimize the risk of bias in the stages of selection, data extraction and quality assessment was executed by two researchers using Cochrane’s Risk of Bias Tool 2 and GRADE. Presented review was registered with OSF (https://osf.io/45tcd, Date registered May 30, 2020) and is currently in progress.
The methodological decisions (protocol) could lead to differences in conclusions and implications from meta-analyses, thus presented article is very important in terms of methodological rigor. Moreover, the present study as authors declared, will extend knowledge derived from previous meta-analyses by analysing individual participant data (IPD). It's hard to find the weaknesses of this work, however there is lack of results and discussion thus the conclusion of this manuscript is a wishful thinking and speculative matter.
I suggest, major revision, mainly regarded to results, discussion and conclusion part of manuscript. In other words, the authors should obtain the results to fulfill general requirements for research manuscript:
- Results: Provide a concise and precise description of the experimental results, their interpretation as well as the experimental conclusions that can be drawn.
- Discussion: Authors should discuss the results and how they can be interpreted in perspective of previous studies and of the working hypotheses. The findings and their implications should be discussed in the broadest context possible and limitations of the work highlighted. Future research directions may also be mentioned. This section may be combined with Results.
- Conclusions: This section is mandatory, and should provide readers with a brief summary of the main conclusions.
Author Response
We thank the reviewer for the encouraging feedback on the study methods.
In the field of individual participant meta-analyses (IPD-MA), it is good practice to publish study protocols outlining the methods that will be used. This enhances the quality of the final analyses as it reduces sources of bias. Study protocols offer the chance to describe the procedures in detail compared to the results paper that have often limited space for the methods section due to restrictive word counts. Furthermore, an IPD-MA is a collaborative project. The published protocol in a special issue can contribute to obtain data from eligible trials that are currently in the publishing process, as authors in the field will be likely to read the review. Therefore, we would highly appreciate the publication of this manuscript ahead of conducting the analyses. Examples for IPD-MA study protocols:
Breedvelt, J.J.F.; Warren, F.C.W.; Brouwer, M.E.B.; Karyotaki, E.; Kuyken, W.; Cuijpers, P.; van Oppen, P. van; Gilbody, S.; Bockting, C.L.H. Individual participant data (IPD) meta-analysis of psychological relapse prevention interventions versus control for patients in remission from depression: a protocol. BMJ Open 2020, 10, e034158.
Lin, J.; Scott, W.; Carpenter, L.; Norton, S.; Domhardt, M.; Baumeister, H.; Mccracken, L.M. Acceptance and commitment therapy for chronic pain : protocol of a systematic review and individual participant data meta-analysis. 2019, 8, 1–10.
Karyotaki, E.; Furukawa, T.A.; Efthimiou, O.; Riper, H.; Cuijpers, P. Guided or self-guided internet-based cognitive-behavioural therapy (iCBT) for depression? Study protocol of an individual participant data network meta-analysis. BMJ Open 2019, 9.
Round 2
Reviewer 1 Report
There needs tone a pilot test done with results. A methodology is not sufficient for a paper.
Reviewer 2 Report
I accept manuscript in present form